# 7-Methylxanthine Inhibits the Formation of Monosodium Urate Crystals by Increasing Its Solubility

**DOI:** 10.3390/biom13121769

**Published:** 2023-12-10

**Authors:** Antonia Costa-Bauza, Felix Grases

**Affiliations:** Laboratory of Renal Lithiasis Research, University Institute of Health Science Research (IUNICS-IdISBa), University of Balearic Islands, Ctra. de Valldemossa km 7.5, 07122 Palma, Spain; antonia.costa@uib.es

**Keywords:** monosodium urate, 7-methylxanthine, gout, solubility enhancement

## Abstract

Gout is characterized by the formation of monosodium urate crystals in peripheral joints. We carried out laboratory studies to investigate the effect of adding nine different methylxanthines and two different methylated uric acid derivatives on the development of these crystals over the course of 96 h in a medium whose composition was similar to that of synovial fluid. Our results showed that 7-methylxanthine reduced or totally prevented crystal formation; 1-methylxanthine, 3-methylxanthine, 7-methyluric acid, and 1,3-dimethyluric acid had weaker effects, and the other molecules had no apparent effect. The presented results indicate that a 7-methylxanthine concentration of about 6 × 10^−5^ M (10 mg/L) prevented the formation of crystals for an initial urate concentration of 1.78 × 10^−3^ M (300 mg/L) in the presence of 0.4 M of Na^+^ for 96 h at 25 °C and a pH of 7.4. We attribute these results to alterations in thermodynamics, not kinetics. Our results suggest that prevention of crystallization in vivo could be achieved by direct oral administration of 7-methylxanthine or other methylxanthines that are metabolized to 7-methylxanthine. For example, the hepatic metabolism of theobromine leads to significant plasma levels of 7-methylxanthine (14% of the initial theobromine concentration) and 3-methylxanthine (6% of the initial theobromine concentration); however, 7-methyluric acid is present at very low concentrations in the plasma. It is important to consider that several of the specific molecules we examined (theobromine, caffeine, theophylline, dyphylline, etophylline, and pentoxifylline) did not directly affect crystallization.

## 1. Introduction

Gout is a rheumatic disease caused by the accumulation of monosodium urate (NaU) crystals in the peripheral joints, especially in the metatarsophalangeal joint in the big toe and other joints in the feet and hands. These needle-shaped crystals cause pain, swelling, and redness in the affected joints. Uric acid (UA) is a product of purine metabolism that occurs as univalent anionic urate at a pH of 7.4, the normal pH of plasma and synovial fluid. Healthy individuals have a serum urate level below 3.6 × 10^−4^ M (6 mg/dL) and a Na^+^ level of about 0.15 M, and NaU crystals do not form under these conditions. However, people who have certain genetic factors, who consume foods that are rich in purines (red meat, viscera, sausages, fish, and shellfish), and who drink alcoholic beverages (especially beer) can accumulate significant amounts of urate, with a serum level up to 1.2 × 10^−3^ M (20 mg/dL) [1].

Sodium urate has a solubility limit of 4 × 10^−4^ M (6.6 mg/dL) of urate in a sodium phosphate buffer with 0.15 M NaCl at a pH of 7.40 and 37 °C [2]. When urate concentration exceeds this level in serum and synovial fluids, it forms fine needle-like NaU crystals that are responsible for the symptoms of gout. The solubility of NaU visibly decreases as the temperature decreases, and its solubility is only 2.2 × 10^−4^ M (3.7 mg/dL) at 26 °C [2]. The lower temperatures of the hands and feet are, therefore, responsible for the formation of crystals in synovial joints [2]. Furthermore, when urine has an excess of UA and a pH below 5.5, UA can crystallize into kidney stones.

One mainstay of treatment for gout is following a strict diet to prevent excessive production of UA. There are also three general strategies for the pharmacological treatment of these patients, as follows:Drugs that reduce the pain from gout attacks, such as non-steroidal anti-inflammatory drugs;Drugs that decrease the production of UA, such as allopurinol and febuxostat;Drugs that increase the urinary excretion of UA, such as probenecid.

Inhibitors of the development of solids within a liquid can be classified into two large categories: (a) inhibitors that are adsorbed at the points of crystalline growth, which usually act at low concentrations, for supersaturation values which are not too high and at which crystal growth does not take place through surface nucleation [3,4], and (b) inhibitors that reduce supersaturation because they are able to achieve some type of reaction (complexation, cluster formation, redox reaction, etc.) with the substance that would otherwise precipitate, which generates a decrease in the latter’s free concentration, meaning that its supersaturation also decreases. Supersaturation is the driving force of crystallization [5]; therefore, its decrease produces an apparent increase in solubility. Inhibitors that modify supersaturation sometimes also act as inhibitors of crystal development.

At this time, there are no available drugs that increase the solubility of NaU in synovial fluids through the formation of aggregates in a solution. Previous research showed that theobromine can form aggregates with UA in a solution and thereby increase the urinary solubility of UA [6]. Hydrogen bonds and π-stacking bonds (aromatic stacking interactions) of UA with theobromine or melamine are responsible for these aggregates [6,7]. A similar mechanism is responsible for the increased solubility of UA in the presence of vitamin C [8]. Nevertheless, theobromine also acts as an inhibitor of UA crystallization through interaction with its crystals, which leads to the UA crystals formed having a different morphology in its presence than in its absence [9]. Moreover, theobromine and some of its metabolites (methylxanthines which lacked a substituent at position 1) contribute globally to reducing the risk of UA stones [10]. In this sense, it has been shown that the consumption of cocoa-derived products, which are rich in theobromine, reduces the risk of crystallization of UA in urine [11,12]. It has also been found that 7-methylxanthine, 3-methylxanthine, and 1-methylxanthine are substances that inhibit xanthine crystallization; so, they could prevent the development of xanthine kidney stones in patients with xanthinuria [13]. The content of theobromine, caffeine, theophylline, and their metabolites in human plasma and urine after consumption of soluble cocoa products have been evaluated [14]. The pharmacokinetic parameters indicated that, in plasma, theobromine is the predominant methylxanthine and that the concentration of 7-methylxanthine is about 13% of that of theobromine, whereas, in urine, 7-methylxanthine is the predominant compound, present at slightly higher levels or even at double concentrations than theobromine [14].

It is important to consider the substantial difference that exists between the crystallization of UA in urine and the formation of NaU in synovial fluid. For example, the formation of UA crystals in renal cavities is generated in a medium with open circulation of urine that is continuously renewed and removed, while synovial fluid, located in the cavities of synovial joints, is an ultrafiltrate from plasma that undergoes a much slower renewal. This implies that the action of classical crystallization inhibitors, which can prevent the formation of crystals for periods of time up to 30–40 or even 60 min, is useful to avoid the formation of crystals inside the kidney but ineffective to prevent crystal formation in synovial joints with limited fluid circulation. Currently, strategies have been proposed to prevent the development of needle-like NaU crystals, stabilizing amorphous NaU. In one of these strategies, the synergistic cooperation of an arginine-rich peptide and copper ions delayed the crystallization of NaU by about 48 h [15,16]. Another strategy involved the addition of substances that form cocrystals with NaU in such a way that the crystals formed would be more soluble (competing binding agents). Thus, the addition of trimethoprim to urate solutions significantly delayed the NaU crystallization times because it yielded a new cocrystal with a solubility greater than that of NaU [17].

The objective of the present study is to identify molecules that increase the solubility (solubility enhancers) of NaU through the formation of aggregates with urate in an artificial medium that resembles synovial fluid.

## 2. Materials and Methods

### 2.1. Reagents

The effects of nine methylxanthines and two methylated UA derivatives on the crystallization of NaU were studied (Figure 1). The following molecules were obtained from Sigma-Aldrich (St. Louis, MO, USA): caffeine (1,3,7-trimethylxanthine), theobromine (3,7-dimethylxanthine), theophylline (1,3-dimethylxanthine), 1-methylxanthine, 3-methylxanthine, 7-methylxanthine, 1,3-dimethyluric acid, 7-methyluric acid, pentoxifylline, dyphylline, and etophylline. A urate stock solution (5.95 × 10^−3^ M = 1 g/L) was prepared by dissolving 0.5 g of UA (Sigma-Aldrich St. Louis, MO, USA) in 0.5 L of water, followed by the addition of NaOH to achieve a final pH close to 9. All stock solutions were prepared using Milli-Q water and were stored at 25 °C, and all the products were of reagent-grade quality.

### 2.2. Sodium Urate Crystallization

Crystallization experiments were performed at room temperature (25 °C) in a solution containing 0.012 M of phosphate buffer, 0.40 M of Na^+^, and an initial urate level of 1.79 × 10^−3^ to 2.84 × 10^−3^ M (300 to 475 mg/L) at a pH of 7.41. The initial urate level and optimal monitoring time of the experiments were evaluated.

The initial urate level for NaU crystallization was stablished considering that, as previously published, at 37 °C NaU does not crystallize spontaneously at urate concentration of 5 × 10^−3^ M (840 mg/L) in the presence of 0.14 M of Na^+^ (normal concentration in plasma and synovial fluid) [18]. It was also taken into account that, over that Na^+^ level, the solubility of NaU is only slightly affected by the increase in Na^+^ concentration, while temperature has a considerable influence on it, since, at 26 °C NaU solubility is approximately 40% lower than that at 37 °C [2]. Therefore, we used a Na^+^ concentration of 0.40 M throughout all the experiments. As can be seen in Figure 2, the solution containing 1.19 × 10^−3^ M (200 mg/L) of initial urate (0.40 M of Na^+^, 0.012 M of phosphate buffer, pH of 7.41, 25 °C) did not crystallize during the observation period, while that containing 1.64 × 10^−3^ M (275 mg/L) of initial urate crystallized slightly. For that reason, the experiments were performed with initial urate levels of 1.79 × 10^−3^ to 2.84 × 10^−3^ M (300 to 475 mg/L). The monitoring time of the experiments was set to 96 h, because, for longer times, only a slight decrease in the amount of urate (*U*_S_) remaining in the supernatant (described below) was observed and also due to the fact that, after that time, a reduction in the urate in the solution was due to its decomposition by the action of atmospheric O_2_ [19].

### 2.3. Study on the Effects of Methylxanthines and Methylated Uric Acid

The study on effect of different compounds on the crystallization of NaU was performed in Falcon 12-well sterile non-treated polystyrene microplates (Corning, Durham, NC, USA), each containing 5 × 10^−3^ L of a crystallizing solution, that were sealed with polyester microplate sealing tape (Corning, Durham, NC, USA) to prevent evaporation. The plates were stored for 96 h at room temperature (25 °C) without stirring, except for experiments which examined the influence of temperature. All the experiments were performed in triplicate, and the results were reported as means with the standard error of the mean.

The effect of different compounds (Figure 1) at 2.75 × 10^−4^ M on the crystallization of NaU was examined using an initial urate concentration of 2.25 × 10^−3^ M (375 mg/L). For these experiments, a solution containing 7.5 × 10^−2^ L of 5.95 × 10^−3^ M (1 g/L) urate, 2 × 10^−2^ L of 0.12 M monosodium phosphate, 3.88 × 10^−2^ L of 2 M sodium chloride, and 3.88 × 10^−2^ L of water was first prepared. After adjustment to a pH of 7.41, 4.315 × 10^−3^ L of this solution was placed into each well, containing 0.685 × 10^−3^ L of water (control) or 0.685 × 10^−3^ L of a 2 × 10^−3^ M solution of the compound under study.

The effects of 7-methylxanthine and 3-methylxanthine were examined using the same procedure, but an appropriate volume of 2 × 10^−3^ M of each compound was added to achieve the desired concentration.

The influence of different initial urate concentrations from 1.79 × 10^−3^ to 2.84 × 10^−3^ M (300 to 475 mg/L) on the effects of 7-methylxanthine (5 × 10^−6^ to 3.5 × 10^−4^ M) was also determined. In this case, different volumes of a 5.95 × 10^−3^ M (1 g/L) urate solution were used. The influence of temperature on the effects of 7-methylxanthine was studied at an initial urate concentration of 2.40 × 10^−3^ M (400 mg/L), with incubation of 12-well plates at 28 °C, 31 °C, or 36 °C for 96 h.

The effects of the different treatments on NaU crystallization were evaluated after 96 h by measuring the amount of urate (*U*_S_) remaining in the solution (described below). Crystallization was qualitatively evaluated by naked eye observations and using scanning electron microscopy (SEM; TM4000 Plus II, Hitachi, Tokyo, Japan). For the SEM measurements, a pipette was used to collect crystals from the bottom of the well; the crystals were placed on a sample holder and fixed with adhesive conductive tape, and the crystals were then dried using paper to absorb any remaining liquid.

The lowest concentration of 7-methylxanthine that totally prevented the formation of NaU crystals after 96 h under different conditions was defined as the minimum effective concentration (*MEC*). This value is analogous to the minimum plasma concentration of a drug needed to produce a desired pharmacologic response.

### 2.4. U_S_ Determination

The concentration of urate remaining in the supernatant (*U*_S_) after the 96 h incubation period was determined using an adaptation of an established colorimetric method in which urate reacts with phosphotungstic acid in the presence of sodium carbonate to produce tungsten blue, whose absorbance at 700 nm after 20 min is proportional to the concentration of urate [20]. Prior to these measurements, it was confirmed that 7-methylxanthine did not interfere in the determination of urate when the 7-methylxanthine concentration was up to 40 times higher than that of urate in the experimental samples. For the measurements of the experimental samples, a 4 × 10^−5^ L aliquot of the supernatant from each well was diluted to 2.50 × 10^−4^ L and then mixed with 5 × 10^−5^ L of CO_3_Na_2_ (0.94 M) and 5 × 10^−5^ L of a 1:5 dilution of the commercial Folin reagent (Sigma-Aldrich, St. Louis, MO, USA) in 96-well plates. A standard curve was established using urate standards at concentrations of 6 × 10^−5^ to 3.6 × 10^−4^ M (10 to 60 mg/L) in 1.92 × 10^−3^ M of phosphate buffer and 6.4 × 10^−2^ M of Na^+^.

## 3. Results

Our measurements of the levels of soluble urate remaining after 96 h indicated that six of the tested compounds had no effect on preventing the crystallization of NaU (Figure 3). However, 7-methylxanthine significantly prevented the formation of crystals, and 1-methylxanthine, 3-methylxanthine, 7-methyluric acid, and 1,3-dimethyluric acid had weaker effects.

We then evaluated the effects of different concentrations and mixtures of 7-methylxanthine and 3-methylxathine to identify possible synergistic effects (Figure 4). The effects of 7-methyluric acid were not studied in more detail since its plasma levels are negligible. We therefore measured the concentration of urate that remained in the solution (*U*_S_) after 96 h at 25 °C, with an initial urate concentration of 2.25 × 10^−3^ M (375 mg/L), following addition of different concentrations and mixtures of 7-methylxanthine and 3-methylxathine. The results show that these two compounds had additive effects and not synergistic effects.

We then evaluated the effect of 7-methylxanthine concentration (0 to 3.5 × 10^−4^ M) on crystallization after 96 h at 25 °C when there were different initial concentrations of urate (1.79 × 10^−3^ to 2.84 × 10^−3^ M (300 to 475 mg/L); Figure 5). These results allowed us to establish the *MEC* value for 7-methylxanthine, i.e., the minimum concentration which totally prevented the formation of crystals over 96 h at 25 °C in the presence of different initial levels of urate (Figure 6). These results indicated that the *MEC* value for 7-methylxanthine corresponding to 1.20 × 10^−3^ M (200 mg/L) of urate (highest level of urate in synovial fluid) would be much lower than 5.0 × 10^−5^ µM.

The effect of 7-methylxanthine preventing the formation of NaU was maintained throughout the observation time of the experiments, as can be seen in Figure 7. Thus, the presence of 1.5 × 10^−4^ M of 7-methylxanthine avoided the formation of NaU crystals for at least 120 h. The fact that urate solutions are unstable, since they decompose due to oxidation [19], has to be considered, and, for this reason, the follow-up was not carried out for longer times.

Temperature significantly increases the solubility of NaU [2]. We therefore examined the influence of temperature on the *MEC* of 7-methylxanthine in the presence of 2.38 × 10^−3^ M (400 mg/L) of initial urate (Figure 8). The results show that, as temperature increased from 25 °C to 36 °C, the *MEC* of 7-methylxanthine decreased from 1.5 × 10^−4^ M to 8.0 × 10^−5^ M.

Finally, we used SEM to observe the NaU crystals that formed in the presence and absence of 7-methylxanthine (Figure 9). These crystals had a similar morphology in the absence of 7-methylxanthine and with two different concentrations of 7-methylxanthine (1.5 × 10^−4^ M and 3.0 × 10^−4^ M).

## 4. Discussion

Our major finding was that 7-methylxanthine and four other molecules to a much lesser extent (1-methylxanthine, 3-methylxanthine, 7-methyluric acid, and 1,3-dimethyluric acid) avoided the formation of NaU crystals. In particular, 7-methylxanthine significantly increased the solubility of NaU in our in vitro samples, and there was no crystallization after 96 h when a sufficient concentration of 7-methylxathine was used. The apparent increase in solubility of NaU is based on the formation of clusters (aggregates) in the solution between the urate ion and 7-methylxanthine, which consequently decrease the supersaturation of NaU in the solution; so, by decreasing the driving force of crystallization, its crystallization is prevented or hindered (thermodynamic effect). Apparently, the solubility of NaU increases, but, in reality, what occurs is a decrease in its supersaturation due to the formation of stable complexes between urate ions and 7-methylxanthine analogous to theobromine–UA complexes [6,7]. We therefore attribute the observed effect to thermodynamics and not kinetics. In fact, our SEM analysis (Figure 9) showed that 7-methylxanthine did not alter the morphology of crystals. This indicates that 7-methylxanthine functioned as a “solubility enhancer”, in that it increased the solubility of NaU. Recent studies reported that other molecules, such as theobromine and citrate, increased the solubility of UA [6,7,8]. It is important to keep in mind that, when assessing the action of a substance in the formation of a crystal, a distinction is made between the formation of the crystal in an open system (i.e., kidney) or in a closed system (i.e., joint). In open systems, small crystals can be eliminated more easily, meaning that the crystallization inhibitors may manifest protecting effects even though a crystal has formed. However, in closed systems, if a crystal forms and the solution is supersaturated, it will always end-up growing. Therefore, in this last case, it is more important and effective to increase the solubility of the product that crystallizes to avoid its development into crystals.

It is important to consider that both homogeneous (spontaneous) and heterogeneous nucleation of NaU is very difficult [18,21]. It has been observed that a concentration of urate above 5.06 × 10^−3^ M (850 mg/L) is necessary for homogeneous nucleation to occur [18]. This fact means that more important than the search for nucleation inhibitors is the decrease in supersaturation, since, when this decreases, the ability to form homogeneous nuclei also decreases.

The effects reported here have some similarities and some differences to the previously reported effects of theobromine on the crystallization of UA [9]. A similarity is that theobromine and 7-methylxanthine effectively inhibit the formation of UA crystals because the complexes formed between these compounds and UA decrease the supersaturation of the UA [6]. Moreover, there is also evidence that theobromine can contribute to the dissolution of existing UA kidney stones [22]. Our studies on 7-methylxanthine, a solubility enhancer which increases the solubility of NaU, allowed us to determine the in vitro concentration of 7-methylxanthine that totally prevents crystallization, even when urate is present in amounts much higher than those typical of synovial fluid. More specifically, we identified the *MEC* values of 7-methylxanthine that totally prevented the formation of NaU crystals over a period of 96 h in the presence of different initial concentrations of urate (Figure 5 and Figure 6).

Our results suggest that a 7-methylxanthine concentration of about 6.0 × 10^−5^ M (10 mg/L) prevented the formation of crystals for an initial urate concentration of 1.78 × 10^−3^ M (300 mg/L) in the presence of 0.40 M of Na^+^ for 96 h at 25 °C and a pH of 7.4. This effective concentration of 7-methylxanthine is 30 times less than the urate concentration, which indicates that 7-methylxanthine exerts its effects at concentrations much lower than that of urate. This suggests that 7-methylxanthine has potential use for the prevention of gout. We found that 3-methylxanthine also prevented crystallization, although it has to be present at a four-fold-greater concentration than 7-methylxanthine. We also found that 7-methyluric acid had a slightly stronger effect than 3-methylxanthine, despite the plasma levels of 7-methyluric acid being negligible. In addition, our results suggest the mechanism underlying the previously reported observations that consumption of coffee [23,24,25] or dark chocolate [26] decreased the risk of gout. It is necessary to consider that, although caffeine itself does not affect the crystallization of NaU, 8% of an oral dose of caffeine (consumed as coffee) is metabolized to 7-methylxanthine [27], and 30% of an oral dose of theobromine (consumed as dark chocolate) is metabolized to 7-methylxanthine [27]. Therefore, the hepatic metabolites of caffeine and theobromine are responsible for the prevention of NaU crystallization. In other words, our results suggest that direct consumption of 7-methylxanthine or one of its precursors (theobromine or caffeine) has potential for preventing the crystallization of NaU and the development of gout.

In a recent study performed with an initial urate concentration of 3 × 10^−2^ M (5000 mg/L) at pH = 9.0, in which NaU crystalized when evaporating the solution in the absence and presence of additives at concentrations ranging from 1 × 10^−3^ M to 7 × 10^−3^ M, it was seen that 1-methyluric acid mainly affected the nucleation of NaU, while 1,3-dimethyluric acid significantly suppressed the growth of NaU spherulites after nucleation [28]. Evidently, the formation of NaU in this study takes place in very different conditions to ours, but their results support the fact of the interaction of additives with urate, attributed to hydrogen bonds.

Importantly, several recent studies have examined the use of 7-methylxanthine for the treatment of myopia [29,30], and other studies documented its chronic and sub-chronic toxicity, genotoxicity, and mutagenicity [31,32,33].

An oral dose of 7-methylxanthine up to 1000 mg/kg of one’s body weight has no toxic effects, indicating that the consumption of 400 mg three times per day is a safe approach for preventing gout.

A study on rabbits reported that the oral administration of 30 mg/kg of body weight of 7-methylxanthine led to a peak serum concentration of about 7.0 × 10^−5^ M, with a half-life of 1 h, and a study on human adults found that oral administration of 400 mg led to a peak serum concentration of 2.0 × 10^−5^ M, with a half-life of 200 min [30]. Considering that the *MEC* of 7-methylxanthine is 6.0 × 10^−5^ M for an initial urate concentration of 1.78 × 10^−3^ M (300 mg/L) and a Na^+^ concentration of 0.40 M (Figure 6), the serum concentration achieved by oral administration of 400 mg of 7-methylxanthine three times per day should be effective in preventing the crystallization of NaU in synovial fluid.

Finally, we must also consider that the consumption of an effective dose of 7-methylxanthine is not associated with adverse effects, although its dose could involve up to three administrations/day, which would be a disadvantage compared to the two drugs currently used. This suggests that 7-methylxanthine or its precursors can be safely combined with other drugs that block the production of UA, such as allopurinol or febuxostat, and may also allow for the use of lower doses of these other drugs to decrease or prevent their adverse effects, which can be substantial. The most important limitation of the paper presented here is that this is a study developed in vitro, so it is necessary to verify these results through the corresponding clinical trials in humans.

## 5. Conclusions

7-methylxanthine avoided the formation of NaU crystals in a medium of composition analogous to that of synovial fluid. The lack of toxicity of this molecule in the amounts that should be used to prevent the formation of NaU deposits in the joints indicates the possibility of using it for the treatment of gout, although it is necessary to demonstrate these effects through the corresponding clinical trials in humans.

## Figures and Tables

**Figure 1 biomolecules-13-01769-f001:**
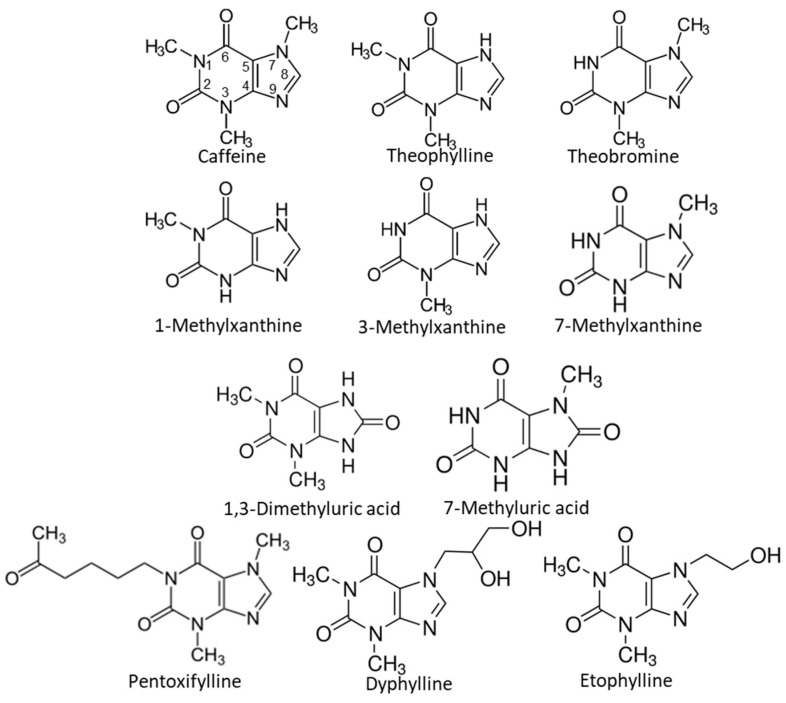
Chemical structures of the nine methylxanthines and the two methylated uric acid derivatives.

**Figure 2 biomolecules-13-01769-f002:**
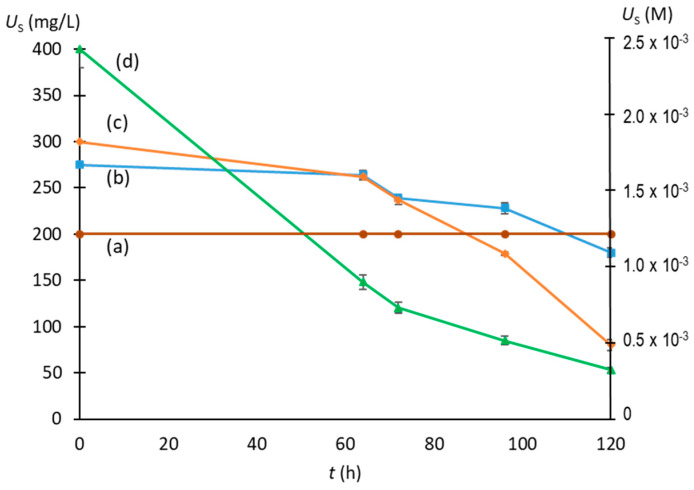
Evolution of concentration of urate remaining in the supernatant (*U*_S_) with time for different initial urate concentrations: (**a**) [urate]_0_ = 1.19 × 10^−3^ M (200 mg/L), (**b**) [urate]_0_ = 1.64 × 10^−3^ M (275 mg/L), (**c**) [urate]_0_ = 1.80 × 10^−3^ M (300 mg/L), and (**d**) [urate]_0_ = 2.38 × 10^−3^ M (400 mg/L). Experimental conditions: 0.40 M of Na^+^, 0.012 M of phosphate buffer, a pH of 7.41, 25 °C. The values are means of triplicates ± standard error of the mean.

**Figure 3 biomolecules-13-01769-f003:**
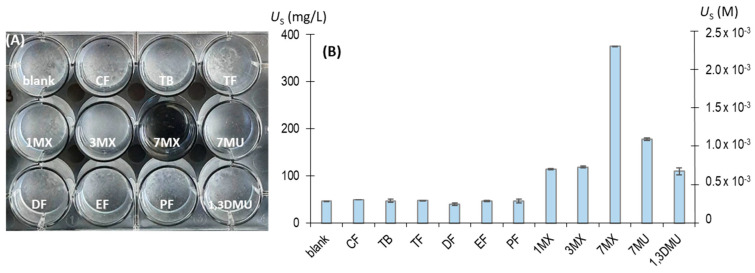
Effects of eleven different molecules (2.75 × 10^−4^ M) on NaU crystallization after 96 h. Experimental conditions: 2.25 × 10^−3^ M (375 mg/L) of initial urate, 0.40 M of Na^+^, 0.012 M of phosphate buffer, pH of 7.41, 25 °C. (**A**) Representative image of a 12-well plate, showing white crystal deposits at the bottoms of all wells except for the one with 7-methylxanthine. (**B**) Concentration of urate remaining in the supernatant (*U*_S_) in the different solutions. blank: no additive; CF: caffeine; TB: theobromine; TF; theophylline; 1MX: 1-methylxanthine; 3MX: 3-methylxanthine; 7MX: 7-methylxanthine; 7MU: 7-methyluric acid; DF: dyphylline; EF: etophylline; PF: pentoxifylline; and 1,3DMU: 1,3-dimethyluric acid. The values are means of triplicates ± standard error of the mean.

**Figure 4 biomolecules-13-01769-f004:**
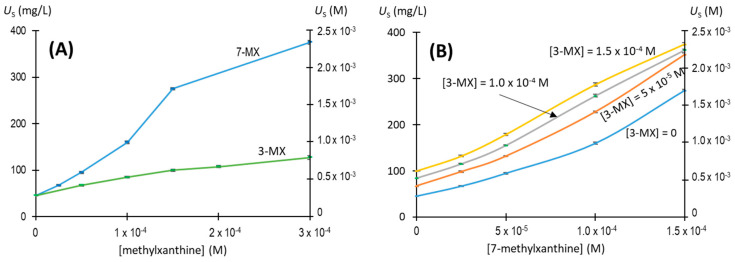
Effects of 7-methylxanthine and 3-methylxantine concentrations and mixtures on NaU crystallization after 96 h. Experimental conditions: 2.25 × 10^−3^ M (375 mg/L) of initial urate, 0.40 M of Na^+^, 0.012 M of phosphate buffer, pH of 7.41, 25 °C. (**A**) Concentration of urate remaining in the supernatant (*U*_S_) when different concentrations of individual molecules were added. (**B**) Concentration of urate remaining in the supernatant (*U*_S_) when different mixtures of both molecules were added. 3-MX: 3-methylxanthine; 7-MX: 7-methylxanthine. The values are means of triplicates ± standard error of the mean.

**Figure 5 biomolecules-13-01769-f005:**
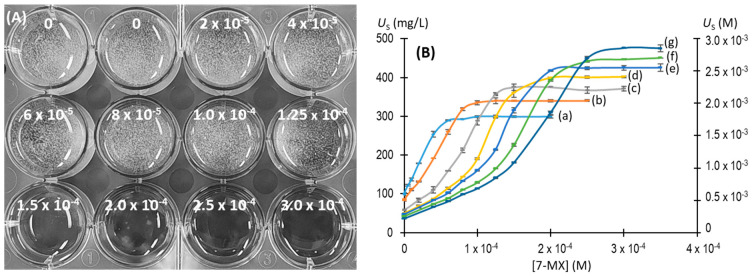
Effects of 7-methylxanthine (7-MX) concentration on NaU crystallization after 96 h. Experimental conditions: 1.79 × 10^−3^ to 2.84 × 10^−3^ M (300 to 475 mg/L) of initial urate, 0.40 M of Na^+^, 0.012 M of phosphate buffer, pH of 7.41, 25 °C. (**A**) Representative image of a 12-well plate for an initial urate concentration of 2.40 × 10^−3^ M (400 mg/L) and a 7-methylxanthine concentration ranging from 2 × 10^−5^ to 3.0 × 10^−4^ M, showing white crystal deposits only at the bottoms of the wells which had low concentrations of 7-methylxanthine (0 to 1.25 × 10^−4^ M). (**B**) Concentration of urate remaining in the supernatant (*U*_S_) when there were different initial concentrations of urate—(a) 1.80 × 10^−3^ M (300 mg/L), (b) 2.04 × 10^−3^ M (340 mg/L), (c) 2.25 × 10^−3^ M (375 mg/L), (d) 2.38 × 10^−3^ M (400 mg/L), (e) 2.54 × 10^−3^ M (425 mg/L), (f) 2.69 × 10^−3^ M (450 mg/L), and (g) 2.84 × 10^−3^ M (475 mg/L)—and 7-methylxanthine (0 to 3.5 × 10^−4^ M). The values are means of triplicates ± standard error of the mean.

**Figure 6 biomolecules-13-01769-f006:**
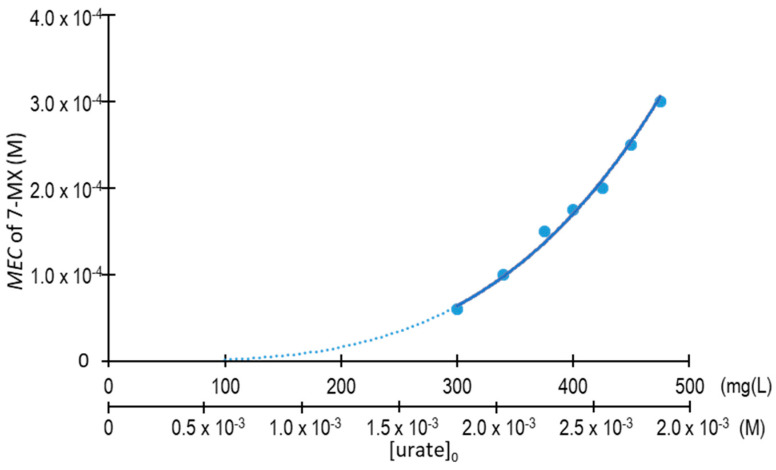
Minimum effective concentration (*MEC*) of 7-methylxanthine that totally prevented the formation of NaU crystals after 96 h. Experimental conditions: 1.79 × 10^−3^ to 2.84 × 10^−3^ M (300 to 475 mg/L) of initial urate, 0.40 M of Na^+^, 0.012 M of phosphate buffer, pH of 7.41, 25 °C. The values have been estimated from Figure 5.

**Figure 7 biomolecules-13-01769-f007:**
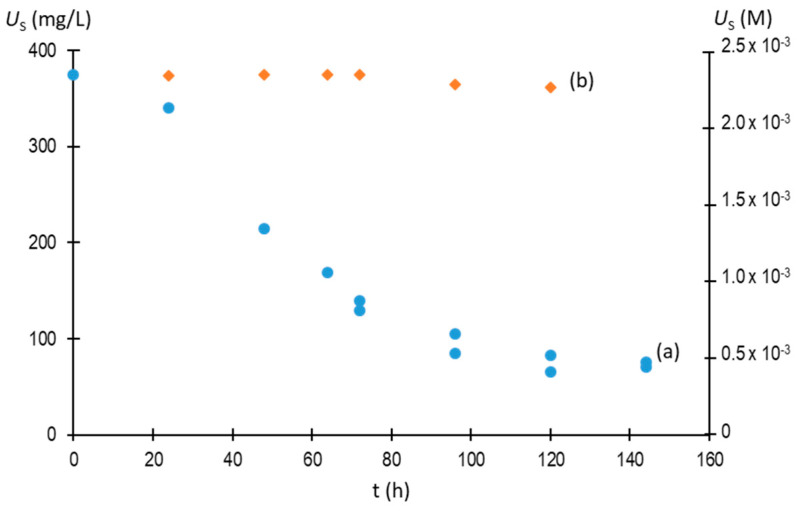
Evolution of concentration of urate remaining in the supernatant (*U*_S_) with time in the absence (**a**) and presence of 1.5 × 10^−4^ M of 7-methylxanthine (**b**). Experimental conditions: 2.25 × 10^−3^ M (375 mg/L) of initial urate, 0.40 M of Na^+^, 0.012 M of phosphate buffer, pH of 7.41, 25 °C.

**Figure 8 biomolecules-13-01769-f008:**
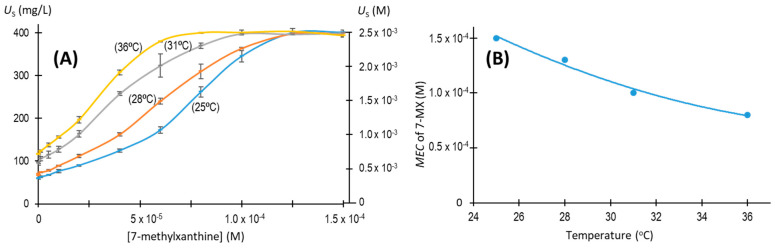
Effects of 7-methylxanthine concentration and temperature on NaU crystallization after 96 h. Experimental conditions: 2.38 × 10^−3^ M (400 mg/L) of initial urate, 0.40 M of Na^+^, 0.012 M of phosphate buffer, pH of 7.41, temperature from 25 °C to 36 °C. (**A**) Concentration of urate remaining in the supernatant (*U*_S_) at different temperatures with different concentrations of 7-methylxanthine. (**B**) Minimum effective concentration (*MEC*) of 7-methylxanthine that totally prevented the formation of NaU crystals at different temperatures. The values in Figure 8A are means of triplicates ± standard error of the mean. The values in Figure 8B have been estimated from Figure 8A.

**Figure 9 biomolecules-13-01769-f009:**
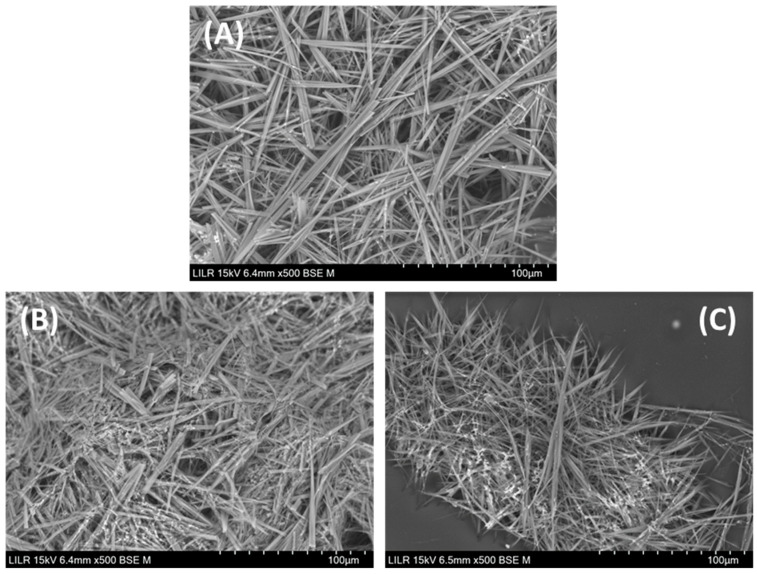
Scanning electron microscopy of NaU crystals in the absence of 7-methylxanthine (**A**), with 1.5 × 10^−4^ M of 7-methylxanthine (**B**), and with 3.0 × 10^−4^ M of 7-methylxanthine (**C**). Experimental conditions: 2.38 × 10^−3^ M (400 mg/L) of initial urate, 0.40 M of Na^+^, 0.012 M of phosphate buffer, pH of 7.41, temperature 25 °C.

## Data Availability

The data that support the findings of this study are available from the corresponding author [F.G.], upon reasonable request.

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
