# Peer review of "7-Methylxanthine Inhibits the Formation of Monosodium Urate Crystals by Increasing Its Solubility"

_biomolecules, 2023, doi:10.3390/biom13121769_

Round 1
Reviewer 1 Report
Comments and Suggestions for Authors
The authors have investigated the inhibition of the formation of monosodium urate crystals by increasing its solubility using nine different methyl xanthines and two different methylated uric acid derivatives. These compounds have been studied as the solubilizers for monosodium urate. Overall, the work is interesting and will be beneficial for the fellow researchers. However, there are several queries need to be addressed before its publication:
Main claim: The main claim of the authors is the use of methyl xanthines and methylated uric acid derivatives as the solubilizers for monosodium urate. Technically, I did not understand this concept. The studied compounds suffer the solubility problems themselves. How could they enhance the solubility of other compounds? What is the main mechanism behind their solubilizing potential? Authors have not performed solubility studies. How could they conclude that the studied compounds are the solubilizers for monosodium urate? Authors are strongly advised to address these queries in the proper scientific manner to satisfy this reviewer.
Abstract: The quantitative results are missing. The authors are suggested to include quantitative results in order to enhance the readability of the manuscript.
Symbols, units, subscripts and superscripts: The authors are advised to present all units in SI system and there should be a space between the physical quantity and unit. All the symbols should be italics and their subscripts or superscripts should be non-italics throughout the manuscript.
Introduction: The main argument is missing in the manuscript. The author’s main objective is to enhance the solubility of monosodium urate using different solubilizers. The importance and significance of solubility measurements are completely missing in the introduction. Authors are advised to add recent literature about the importance of solubility and different solubility approaches. The authors can consult the following articles to make this manuscript more useful to the readers, however they are not restricted to them:
J. Mol. Liq. 255: 43-50 (2018); Molecules 24: E2807 (2019); J. Mol. Liq. 307: E112970 (2020); J. Mol. Liq. 348: E118057 (2022); Molecules 27: E1437 (2022).
Results and discussion: Please compare your results with previous studies and mention clearly how your work is important in comparison to already been reported.
Authors are advised to include the main limitation of work at the end of discussion section and just before the conclusion.
Figure 2: Remove the gridlines from the figure. Kindly add error bars and number of replicates in the figure.
Figure 3B: Remove the gridlines from the figure. Kindly add error bars and number of replicates in the figure.
Figure 4: Remove the gridlines from the figure. Kindly add error bars and number of replicates in the figure.
Figure 5B: Remove the gridlines from the figure. Kindly add error bars and number of replicates in the figure.
Figure 6: Remove the gridlines from the figure. Kindly add error bars and number of replicates in the figure.
Figure 7: There is no trend in the data. Which trend is followed by the present data. Remove the gridlines from the figure. Kindly add error bars and number of replicates in the figure.
Figure 8: Remove the gridlines from the figure. Kindly add error bars and number of replicates in the figure.
Avoid abbreviations before giving their explanation in the abstract, text, table, and figure.
Conclusion: The conclusion should be concise and to the point indicating the application of the work.
References: The authors are strongly advised to follow the target journal guidelines for the preparation of references.
Comments on the Quality of English Language
Minor editing of language is required.
Author Response
Please, see the attachment

Reviewer 2 Report
Comments and Suggestions for Authors
In this manuscript the author proposed the study was designed to explore substances that increase the solubility of sodium urate and artificial solution mimicking synovial fluid. Bottom line they want to introduce an agent for treatment of gout. The results showed that among the different molecules 7-methylxanthine reduced the uric crystal formation. I have the following comments: 1) In order to be sure that the compound can be used in human, the bioavailability and pharmacodynamic of material should be tested with single dose and multiple dose in normal subjects and in patients with gout. this is important since the study was not performed in patients suffering with gout and the safety of the drug has not been tested. 2) You should also describe what are the routes of excretion of metabolites of 7-methylxanthine. 3) I understand that this agent increases the solubility of sodium urate in the4 synovial fluid, would you comment what would be the advantage of this drug compared to urate lowering agents including allopurinol and febuxostat 4) The agent might have a disadvantage over urate lowering drug because it should be administered three times a day to be affective against prevention of urate crystallization according to the discussion on page 10 fourth paragraph.
Author Response
Please, see the attachment

Round 2
Reviewer 1 Report
Comments and Suggestions for Authors
Authors have addressed the previous concerns. Revised manuscript is suitable for publication in its present form.
Comments on the Quality of English LanguageMinor editing of the language is required.